# Bibliometric Analysis and Review of Deep Learning-Based Crack Detection Literature Published between 2010 and 2022

Luqman Ali [1,2,3] , Fady Alnajjar [1,3] , Wasif Khan [1] , Mohamed Adel Serhani [4] and Hamad Al Jassmi [2,5,*]

1   Department of Computer Science and Software Engineering, College of Information Technology, United Arab Emirates University (UAEU), Al Ain 15551, United Arab Emirates; 201990024@uaeu.ac.ae (L.A.); fady.alnajjar@uaeu.ac.ae (F.A.); 201990025@uaeu.ac.ae (W.K.)
2   Emirates Center for Mobility Research, United Arab Emirates University (UAEU), Al Ain 15551, United Arab Emirates
3   AI and Robotics Lab (Air-Lab), United Arab Emirates University (UAEU), Al Ain 15551, United Arab Emirates
4   Department of Information Systems and Security, College of Information Technology, United Arab Emirates University (UAEU), Al Ain 15551, United Arab Emirates; serhanim@uaeu.ac.ae
5   Department of Civil Engineering, College of Engineering, United Arab Emirates University (UAEU), Al Ain 15551, United Arab Emirates
*   Correspondence: h.aljasmi@uaeu.ac.ae

**Abstract:** The use of deep learning (DL) in civil inspection, especially in crack detection, has increased over the past years to ensure long-term structural safety and integrity. To achieve a better understanding of the research work on crack detection using DL approaches, this paper aims to provide a bibliometric analysis and review of the current literature on DL-based crack detection published between 2010 and 2022. The search from Web of Science (WoS) and Scopus, two widely accepted bibliographic databases, resulted in 165 articles published in top journals and conferences, showing the rapid increase in publications in this area since 2018. The evolution and state-of-the-art approaches to crack detection using deep learning are reviewed and analyzed based on datasets, network architecture, domain, and performance of each study. Overall, this review article stands as a reference for researchers working in the field of crack detection using deep learning techniques to achieve optimal precision and computational efficiency performance in light of electing the most effective combination of dataset characteristics and network architecture for each domain. Finally, the challenges, gaps, and future directions are provided to researchers to explore various solutions pertaining to (a) automatic recognition of crack type and severity, (b) dataset availability and suitability, (c) efficient data preprocessing techniques, (d) automatic labeling approaches for crack detection, (e) parameter tuning and optimization, (f) using 3D images and data fusion, (g) real-time crack detection, and (h) increasing segmentation accuracy at the pixel level.

**Keywords:** crack detection; civil inspection; deep learning; literature review; bibliometric analysis; vision-based inspection

## 1. Introduction

Cracks are the initial signs of degradation of any civil infrastructure, which appear due to various reasons, such as structural foundation shifting, shrinkage and expansion, unbalanced blend, swollen soil, overloading, natural hazards, manmade disasters, and so on. Crack detection and localization tasks can be performed by manually gathering information, i.e., visually inspecting and assessing the structure by subjective human experts or automatically [1]. Manual inspection techniques are laborious, time-consuming, inspector dependent, and easily vulnerable to the perspicacity of the inspector. To overcome all the problems associated with manual assessment, automatic inspection techniques provide an efficient solution that decreases subjectivity and can be used as an alternative to the human eye [2]. Additionally, the rate of designing and utilizing the computer

vision-based civil infrastructure inspection and monitoring system has dramatically increased [3]. Developing an efficient vision-based automated pavement crack detection system has always been challenging due to the low contrast between the pavement surface and cracks, irregular size and random shape of cracks, the intensity of variation in the image, and multi-texture and shadowing in the images [4]. Computer vision approaches include acquisition and analysis to find and classify the crack regions [5]. The results from these approaches are used to (a) replace the manual detection and quantification of cracks, (b) provide a reference for crack detection for future structure maintenance, and (c) enable the automatic repair of crack regions by providing accurate positioning of the crack regions. These approaches can be categorized into image-processing-based and machine learning approaches. Image processing methods for crack detection include edge detection [6], thresholding and segmentation [7], region growing [8], and peculation-based techniques [9]. In image-processing-based approaches, the local information-based models use various filters, such as the morphological filter [10], statistical filter [11], 2D matched filter [12], median filter, multi-scale line filter with the Hessian matrix [13], and so on. The main issue with image-processing-based techniques is the poor continuity and low contrast between neighbouring crack pixels. Due to the evolution of machine learning, automated inspection has reached an inevitable peak of advancement. Machine learning approaches can learn deep features and perform statistical inference without tuning parameters manually, as in other prior approaches [14]. Traditional machine learning approaches for pavement crack detection consist of data acquisition, feature extraction, and classification. In data acquisition, high-resolution images of the target structure are acquired and preprocessed to remove noise and other forms of deformities. In feature extraction, the acquired features are given to classifiers to determine their class. Various reviews have been proposed on ML-based crack detection algorithms [3,15], which conclude that if the extracted features do not represent the actual cracks, the classifier may not yield accurate results. Moreover, these ML-based approaches cannot generate semantic features of the cracks. The capability and robustness of traditional approaches have been greatly extended by deep learning techniques [16]. In DL, the term "deep" refers to a large number of layers present between the input and the output layers [17–19]. DL models are different from traditional machine learning approaches in that they are capable of learning the representations of the data without introducing any hand-crafted rules or knowledge and have shown great performance in solving the crack detection problem [1]. Deep learning has the capability to classify, i.e., labeling the image patch as crack or non-crack; localize, i.e., determining the location of the crack by drawing a bounding box around the crack region; and segment, i.e., segmenting the image pixels into crack and non-crack pixels. A sample representation of the output of the DL model is shown in Figure 1. Recently, various DL-based crack detection studies were proposed in research environments. Therefore, to provide the researchers with sufficient knowledge of available literature and maintain its intensity, a detailed study of deep learning-based automatic approaches for crack detection in civil structure is carried out in this article. The main contributions of the articles are as follows:

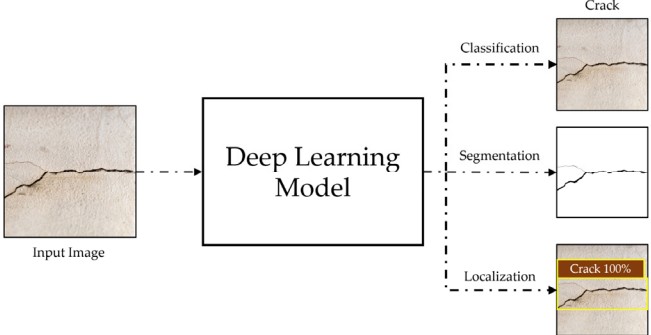

**Figure 1.** Representation of the output of the deep learning model.

- The article performs bibliometric analysis and a comprehensive review of the most recent research articles from the last 11 years.
- In the comprehensive review, the relative finding of the latest studies in DL-based crack detection is explained, and analysis is performed based on various factors, such as the studied domain, deep learning architecture, architecture performance, and datasets used.
- The paper provides an overview of DL-based crack detection approaches and categorizes them to simplify the current literature for better understanding and better analysis of trends, gaps, and challenges in the field. The paper also provides future directions to the readers to address the current challenges. The articles can be used as a reference that will work for the researchers working in the field.

The remainder of this paper is organized as follows. Materials and methods are described in Section 2. Section 3 summarizes DL-based crack detection approaches. Discussion and conclusions are presented in Section 4. Finally, future works are discussed in Section 6.

*Research Questions*

The main questions that are addressed in this article are as follows:

1. Which datasets have been used for crack detection?
2. What is the impact of dataset characteristics on classification performance?
3. What are the different application domains in crack detection?
4. Which deep learning algorithms have been used for crack detection?
5. What are the gaps and challenges of deep learning for crack detection?
6. What are the future research opportunities?

## 2. Materials and Methods

The research methodology adopted in the proposed work consists of three main steps, as depicted in Figure 2.

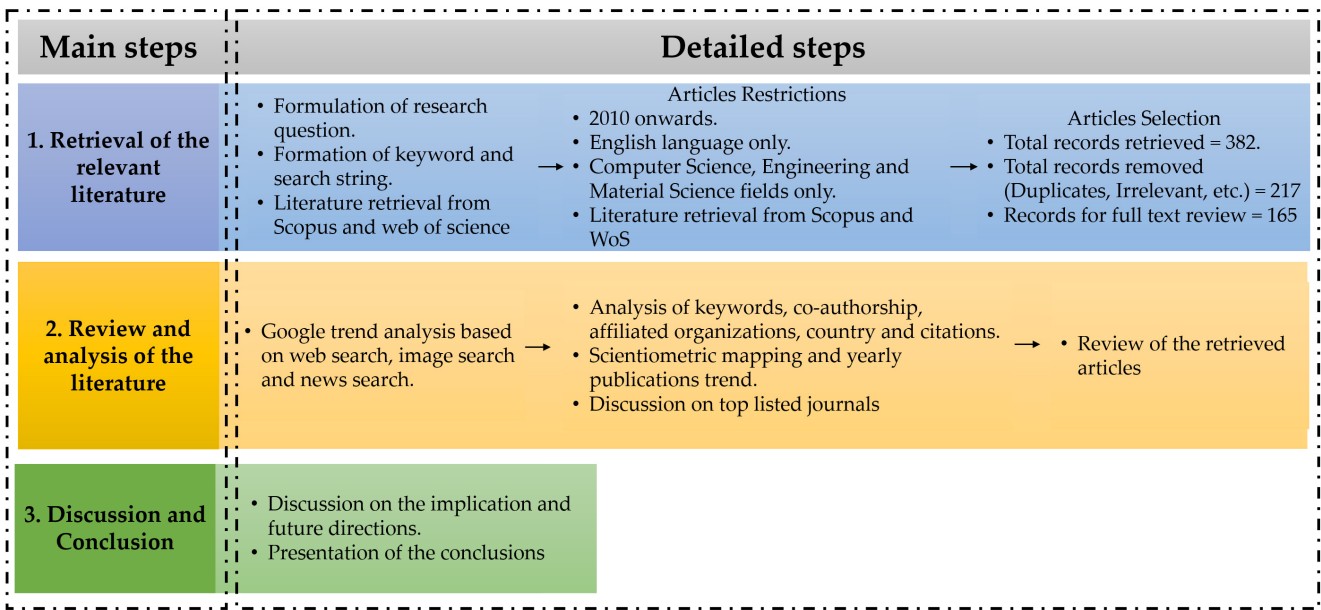

**Figure 2.** Methodology of the current work.

- Retrieval of relevant literature.
- Review and analysis of the retrieved literature.
- Discussion and conclusion.

The first step involves retrieving relevant literature from the Scopus and Web of Science (WoS) search engines. In the second step, a different analysis is performed, which includes analyses of keywords, affiliated organizations, citations, and the article's country of origin. The keyword scientometric mapping, listing of the top articles, and the yearly publication trend of the retrieved articles are also observed in this section. Google trend analysis based on the web, image, and news searches is also performed to visualize the search trend and global focus of the keywords and terms in the proposed study. Furthermore, a detailed review of DL-based crack detection approaches for civil infrastructure monitoring and maintenance is presented. In the last section, discussions, conclusions, and implications are given, and the proposed work is concluded by presenting the future work.

*Literature Retrieval*

In the current study, the literature is retrieved from Scopus and WoS search engines. In the beginning, various research questions are formulated to identify the objectives of the study. The relevant literature is retrieved from the search engines based on their keywords and semantic search strings. The search strings are formed by using the keywords used in previous relevant literature reviews [5,18,19] and conceptual terms that are comparable. The keywords used for searching relevant papers include "crack detection" "deep learning", "crack detection" "Machine Learning", "crack detection" "road" "pavement" "concrete", "crack detection" "manual inspection", and "condition assessment" "computer vision" joined by Boolean operators "OR" as depicted in Table 1. The total number of articles that use the search string keywords in the title or abstract is 523 from Scopus and 367 from the Web of Science. Various restrictions are applied to keep relevant articles related to the current study theme. In the Scopus search engine, the articles are restricted to engineering, computer science, and material science. Similarly, in WoS, restrictions are imposed differently, and the subject areas include civil engineering, construction building technology, computer science interdisciplinary applications, engineering multidisciplinary, computer science artificial intelligence, transportation, materials science multidisciplinary, transportation science technology, computer science information systems, robotics, and imaging science photographic technology. A few more restrictions, as shown in Table 2, are applied to the retrieved articles; the period of publication is restricted from 2010 to 2021 (Criterion 1); the language is restricted to the English language only (Criterion 2); only journals and conferences are considered in the article type (Criterion 3). Conferences and journals with at least five publications in their outlets regarding the topic of interest are included (Criterion 4). This comprehensive search results in 382 articles from both search engines. A total of 99 duplicates found in both the search engines are removed, and a total of 239 articles are selected after applying Criterion 5 (articles related to deep learning only). At the end, the title, abstract, and full paper screening is performed, and 165 relevant articles are selected for the current work.

**Table 1.** The search string, refinements, and results of the Scopus and WoS repositories.

| Search Engine | String and Refinement | Results |
|---|---|---|
| **Scopus** | (TITLE-ABS-KEY ("crack detection" "deep learning") OR TITLE-ABS-KEY ("crack detection" "MachineLearning") OR TITLE-ABS-KEY ("crack detection" "road" "pavement" "concrete") OR TITLE-ABS-KEY ("crack detection" "manual inspection") OR TITLE-ABS-KEY ("condition assessment" "computer vision")) AND PUBYEAR > 2009 AND (LIMIT-TO (SUBJAREA, "ENGI") OR LIMIT-TO (SUBJAREA, "COMP") OR LIMIT-TO (SUBJAREA, "MATE")) AND (LIMIT-TO (DOCTYPE, "ar") OR LIMIT-TO (DOCTYPE, "cp")) AND (LIMIT-TO (LANGUAGE, "English")) | 523 506 219 219 |

**Table 1.** *Cont.*

| Search Engine | String and Refinement | Results |
|---|---|---|
| **WoS** | TOPIC: ("crack detection" "deep learning") OR TOPIC: ("crack detection" "Machine Learning") OR TOPIC: ("crack detection" "road" "pavement" "concrete") OR TOPIC: ("crack detection" "manual inspection") OR TOPIC: ("condition assessment" "computer vision")<br>Timespan: 2010–2021.<br>Indexes: SCI-EXPANDED, SSCI, A&HCI, CPCI-S, CPCI-SSH, BKCI-S, BKCI-SSH, ESCI.<br>Refined by: WEB OF SCIENCE CATEGORIES: (ENGINEERING CIVIL OR CONSTRUCTION BUILDING TECHNOLOGY OR COMPUTER SCIENCE INTERDISCIPLINARY APPLICATIONS OR ENGINEERING MULTIDISCIPLINARY OR COMPUTER SCIENCE ARTIFICIAL INTELLIGENCE OR TRANSPORTATION OR MATERIALS SCIENCE MULTIDISCIPLINARY OR TRANSPORTATION SCIENCE TECHNOLOGY OR COMPUTER SCIENCE INFORMATION SYSTEMS OR ROBOTICS OR IMAGING SCIENCE PHOTOGRAPHIC TECHNOLOGY)<br>AND DOCUMENT TYPES: (ARTICLE OR PROCEEDINGS PAPER)<br>AND LANGUAGES: (ENGLISH) | 367<br>292<br>163<br>163 |

**Sum of the papers = 382**
**Duplicates = 99**
**Remaining 283**
**After applying Criterion 5 = 239**
**After abstract screening = 165**
**After full paper screening = 165**
**Total = 165**

**Table 2.** Inclusion and exclusion criteria followed to extract relevant studies.

| Criterion | Inclusion | Exclusion |
|---|---|---|
| C1: (Publication Year) | 2010–2021 | Paper published before 2010 |
| C2: Language | English | Other languages such as Chinese, German |
| C3: Article Type | Journal and conferences | Books, conference review etc. |
| C4: Number of articles in the outlet | At least 5 publications | Less than five articles |
| C5: Deep Learning | Based on DL | Not based on DL |

## 3. Bibliometric Analysis of the Retrieved Literature

The Google trend analysis, literature synthesis and bibliometric analysis of the retrieved articles are discussed below.

### 3.1. Google Trends Analysis

Google trends analysis of the keywords is performed to highlight the global search, and attention is given to the considered keywords in the current work. The comparison is performed based on three keywords: crack detection, convolutional neural network, and deep learning. Figure 3 shows the web search, news search, and image search trends plotted for the years 2010 to 2021. The web-based search for the keywords is shown in Figure 3a. The graph shows that the keyword "deep learning" is most searched, followed by "Convolutional Neural Network" and "crack detection". A similar trend is observed in the news search, as shown in Figure 3b, where deep learning dominates the other search items. Few searches are also performed for "Convolutional Neural Network" and "crack detection", but the magnitude is negligible as compared to deep learning. Overall, the news search in comparison with web-based search is very limited. When it comes to image-based search, a similar pattern is observed, and deep learning is the top topic, followed by "Convolutional Neural Network" and "crack detection", as shown in Figure 3c.

### 3.2. Literature Synthesis

In the current study, bibliometric analysis is performed to investigate the retrieved articles. The main steps followed in the literature synthesis are discussed in detail below.

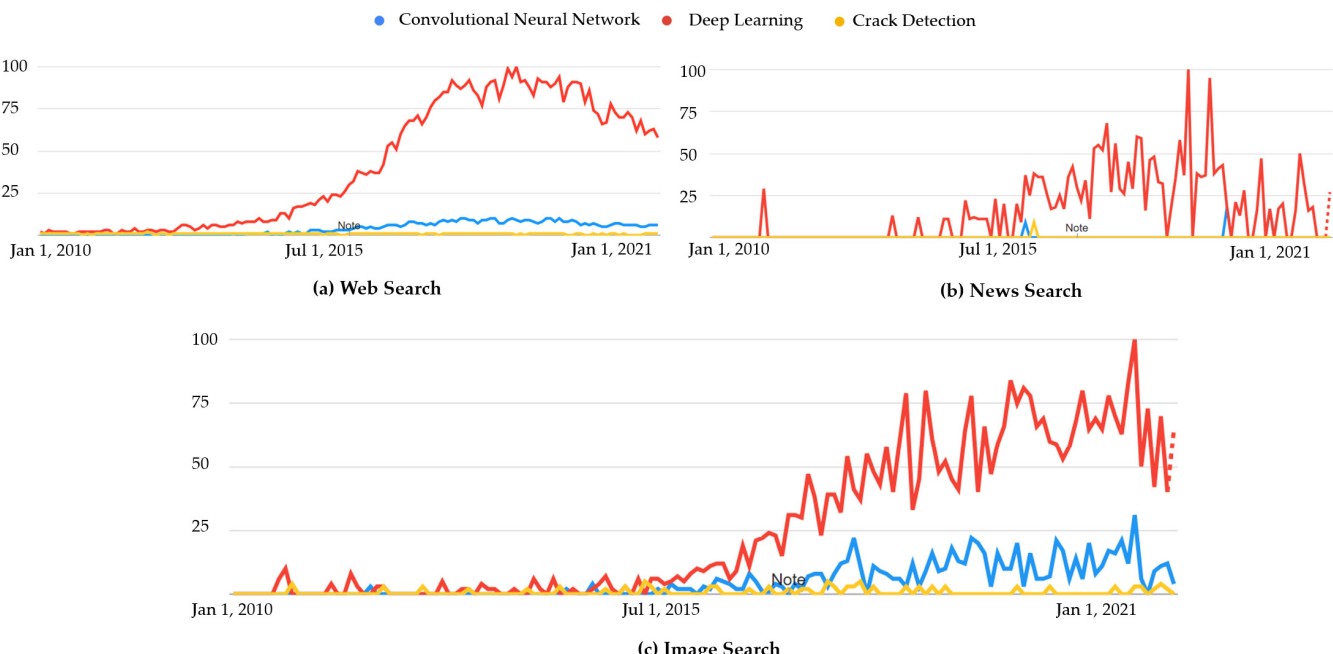

**(a) Web Search**

**(b) News Search**

**(c) Image Search**

**Figure 3.** Google trend analysis of the proposed work keywords. (**a**) Web Search, (**b**) News Search, (**c**) Image Search.

### 3.2.1. Retrieved Articles Classification

The retrieved articles are distributed into three various domains, i.e., computer science, engineering, and material sciences. Most of the articles selected in the current study belong to the field of engineering, followed by computer science and material science, respectively. The statistics in Figure 4 shows that material science is the least explored field, and less than 20% of the retrieved articles belong to this domain. On the other hand, most of the retrieved articles belong to the technology-focused field, i.e., the engineering domain followed by computer science. This also shows that the research environment is focusing on the adoption of technologies and computer applications for civil infrastructure condition assessment problems.

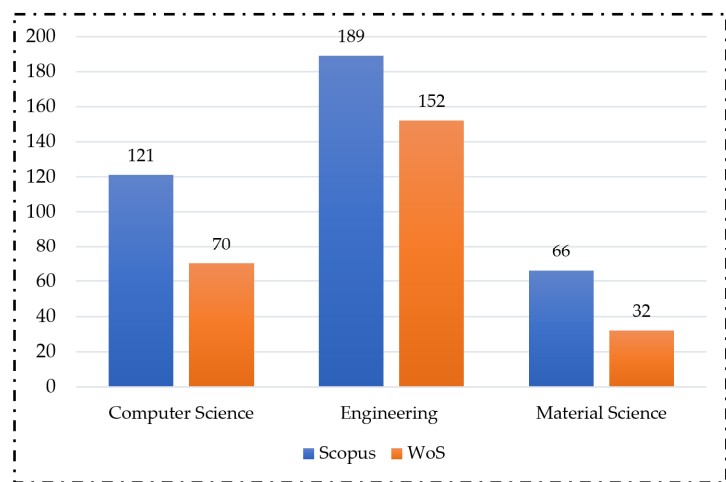

**Figure 4.** Articles categorization (Scopus vs WoS).

### 3.2.2. Publication Trend (Yearly)

The publication trend of the retrieved studies from both repositories is shown in Figure 5. The three different colour graphs represent the studies from Scopus and WoS

repositories and the duplicates in both. As shown in the figure, there has been an increase in the number of publications over the five years, where 90% of the publications have been published in recent years. This highlights the fact that the focus on crack detection using deep learning techniques was increased recently. The nascency of the current study theme can also be verified by the fact that there is a smaller number of publications in the years 2010 to 2015.

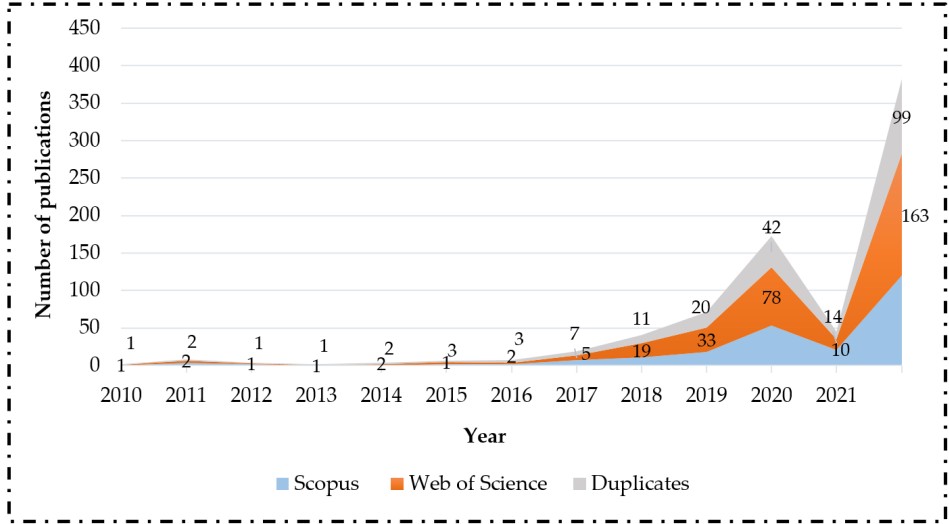

**Figure 5.** Yearly distribution of the retrieved articles.

### 3.2.3. Article Types

After analyzing the yearly publication trend of the retrieved articles, the article's type is examined. In the current study, journal articles and conferences are selected as they provide organized and quality research in comparison with other types of research outputs. The quality of all the retrieved articles fulfils the modern systematic literature review requirements. The articles retrieved from the WoS repository consist of 9% conferences and 91% articles, while in the case of the Scopus repository, the conferences contribute 33%, while the journal articles contribute 67% of the overall articles. In comparison with WoS, the Scopus repository shows a greater tendency toward index conference papers.

### 3.2.4. Top Sources

The sources of the retrieved articles are investigated. The top sources with the most retrieved articles are shown in Table 3. In the inclusion criterion, the outlets in both repositories with a minimum number of three articles published are considered. In the case of the Scopus repository, the top three contributors are the *ACM International Conference Proceeding Series*, *Proceedings of SPIE The International Society for Optical Engineering*, and *Advances in Structure Engineering*. Similarly, for the WoS repository, the top sources are *Automation in Construction*, *Computer-aided Civil and Infrastructure Engineering*, and *IEEE Access*. Amongst the top sources, two outlets, i.e., *Automation in Construction* and *Computer-aided Civil and Infrastructure Engineering*, are common in both search engines. In the top sources, most of the Scopus repository sources are conferences, while the WoS repository sources are mostly journal articles. Overall, the outlets such as *Automation in Construction*, *Computer-aided Civil and Infrastructure Engineering*, *IEEE Access*, *Construction and Building Materials*, *ACM International Conference Proceeding Series*, *Journal of Computing in Civil Engineering*, and *Structural Health Monitoring* contribute to more than half of the retrieved articles.

**Table 3.** Top Scopus and WoS contributors.

| S/No | Scopus Source Name | Documents | Citations | S/No | Web of Science Source Name | Documents | Citations |
|---|---|---|---|---|---|---|---|
| 1 | *Automation in Construction* | 3 | 14 | 12 | *Automation in Construction* | 16 | 328 |
| 2 | *Proceedings of SPIE The International Society for Optical Engineering* | 5 | 45 | 13 | *Computer-aided Civil and Infrastructure Engineering* | 12 | 918 |
| 3 | *SHMII 2019—Conference Proceedings* | 3 | 6 | 14 | *IEEE Access* | 12 | 15 |
| 4 | *Computer-aided Civil and Infrastructure Engineering* | 3 | 955 | 15 | *Journal of Computing in Civil Engineering* | 7 | 86 |
| 5 | *IEEE Region 10 Annual International Conference, Proceedings/TENCON* | 3 | 4 | 16 | *Structural Health Monitoring* | 7 | 26 |
| 6 | *ACM International Conference Proceeding Series* | **7** | 8 | 17 | *Construction and Building Materials* | 8 | 326 |
| 7 | *International Archives of the Photogrammetry, Remote Sensing and Spatial Information Sciences—ISPRS Archives* | 3 | 15 | 18 | *International journal of pavement engineering* | 3 | 6 |
| 8 | *Advances in Structure Engineering* | 4 | 13 | 19 | *Applied Science-Basel* | 7 | 22 |
| 9 | *Applied Sciences Switzerland* | 3 | 10 | 20 | *Materials* | 3 | 4 |
| 10 | *IOP Conference Series: Materials Science and Engineering* | 3 | 1 | 21 | *Structure Control Health Monitoring* | 5 | 103 |
| 11 | *Journal of Performance of Constructed Facilities* | 2 | 7 | 22 | *Journal of Transportation Engineering, Part B: Pavements* | 2 | 4 |

### 3.2.5. Keyword Analysis and Scientometric Mapping

In the current study, the keyword analysis and scientometric mapping of the retrieved studies are performed by using VOS viewer. It is a literature analysis software that is used in various fields to visualize the relationship between authors, fields, authors, organizations, countries, and keywords. The software is mostly used for quantitative analysis and produces the results in a well-organized, meaningful, and easily interpretable way. The co-occurrence analysis for the keywords of the retrieved articles is performed by using VOS viewer, and the results are tabulated in Table 4. From the table, 666 keywords are retrieved from the Scopus repository, which is then restricted to 39 after applying the minimum occurrence criterion. For both repositories, the minimum occurrence criterion is the existence of the keyword in at least five articles. From the WoS, 346 keywords are retrieved, which are then restricted to 32 after applying the minimum inclusion criterion.

**Table 4.** Keyword's co-occurrence analysis.

| Assessment | Consideration | Scopus Outcomes | WoS Outcomes |
|---|---|---|---|
| Analysis Type | Co-occurrence | 666 | 346 |
| Counting method | full counting | | - |
| Units of analysis | all keywords | | - |
| Minimum Occurrence | 5 for Scopus, 5 for WoS | 39 | 32 |

In addition to the keywords co-occurrence analysis, the top retrieved keywords for both the repositories are shown in Table 5. In the Scopus search engine, the top keyword is "Crack Detection", which appears in 57 articles, corresponding to nearly 22.5% of the top keywords. "Deep learning" is the second top keyword with 55 appearances and a 21.6% share. Both the crack detection and deep learning keywords are used in search strings; however, the contribution of both keywords shows the lack of literature on both keywords. All the remaining keywords have less than 11% contribution individually. In WoS, both keywords "Crack detection" and "Deep learning" lead the list with an individual share of 30%, and their appearances are 53 and 54 times, respectively. The top two keywords for both the search engines are the same, regardless of their ranks. "Damage detection"is the third-ranked keyword in the WoS repository, with 29 appearances (16.6%).

**Table 5.** Top 10 keywords.

| Keywords | Scopus Count | Web of Science Count |
|---|---|---|
| Crack Detection | 57 | 53 |
| Deep Learning | 55 | 54 |
| Convolution | 26 | - |
| Convolutional Neural Network | 20 | 14 |
| Convolutional Neural Networks | 20 | 6 |
| Deep Neural Networks | 17 | - |
| Neural Networks | 16 | 8 |
| Structure Health Monitoring | 15 | 5 |
| Damage detection | 13 | 29 |
| Concretes | 15 | 6 |

The scientometric mapping for the keyword of both Scopus and WoS repositories is shown in Figure 6a,b. The inclusion criterion is the 10 and 5 keyword appearances for the Scopus and WoS repositories. In the scientometric analysis of the Scopus search engine as shown in Figure 6a, the top keywords highlighted as large bubbles are "crack detection", "deep learning", "convolutional neural networks", "convolution" and "deep neural networks". The large bubbles represent the high prevalence of the keywords in the retrieved literature. Similarly, in the scientometric mapping of WoS, "crack detection", "deep learning", Convolutional neural network", and "damage detection" are the significant keywords, as shown in Figure 6b. The mapping clearly shows that crack detection using deep learning algorithms is increasingly targeted in published articles as compared to other keywords considered in the current study.

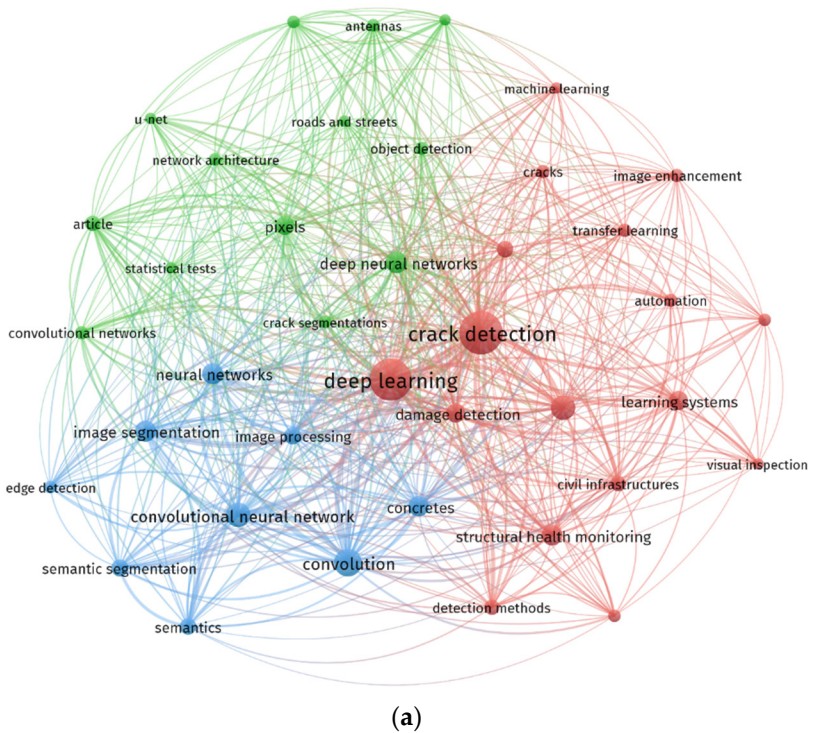

(**a**)

**Figure 6.** *Cont*.

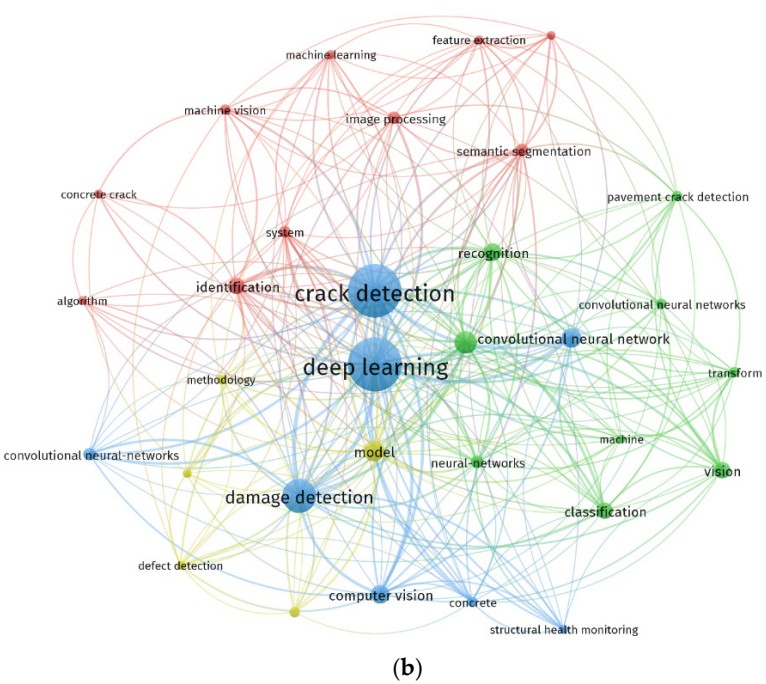

(**b**)

**Figure 6.** Scientometric mapping of Scopus and WoS keywords. (**a**) Scopus keyword mapping, (**b**) WoS keyword mapping.

### 3.2.6. Co-Authorship Analysis

The co-authorship analysis of the retrieved article as shown in Table 6 is performed using a VOS viewer. The inclusion criterion for both search engines is set to two publications by an author. In Scopus articles, out of 227 authors, only 24 authors fulfil the criterion and are considered top contributors to the repository. On the other hand, WoS's top contributors include 45 authors out of 331.

**Table 6.** Co-authorship Analysis.

| Assessment | Consideration | Scopus Results | WoS Results |
| --- | --- | --- | --- |
| Analysis Type | Co-authorship | 227 | 331 |
| Method of counting | Full counting | - | - |
| Analysis units | Authors | - | - |
| Minimum Occurrence | Two for Scopus, two forWoS | 24 | 45 |

### 3.2.7. Organizational Affiliation Analysis

The organizational affiliation of the authors is analyzed by setting the inclusion criterion of two articles per organization as shown in Table 7. In the articles retrieved from Scopus, a total of 122 organizations are affiliated, of which four organizations fulfill the inclusion criterion. Similarly, for the WoS, 30 organizations out of 115 are shortlisted based on the inclusion criterion.

**Table 7.** Organizational affiliation analysis.

| Assessment | Consideration | Scopus Results | WoS Results |
| --- | --- | --- | --- |
| Type of analysis | Co-authorship | 122 | 115 |
| Counting method | Full counting | - | - |
| Units of analysis | Organizations | - | - |
| Minimum Occurrence | 2 for Scopus, 2 for WoS | 4 | 30 |

### 3.2.8. Origin Country of the Articles and Citation Analysis

In the retrieved articles, the contribution of various countries in terms of the number of contributed documents is also investigated as shown in Table 8.

**Table 8.** Origin countries analysis.

| Assessment | Consideration | Scopus Results | WoS Results |
|---|---|---|---|
| Type of analysis | Co-authorship | 22 | 18 |
| Counting method | full counting | - | - |
| Units of analysis | Countries | - | - |
| Minimum Occurrence | 3 for Scopus, 3 for WoS | 5 | 7 |

The minimum inclusion criterion of three documents per country is kept for both Scopus and WoS search engines. In the retrieved articles, 22 countries contributed to the Scopus, of which only 5 countries satisfy the minimum inclusion criterion. In WoS, a total of 18 countries contributed, and 7 were shortlisted after applying the limits. The number of documents and citations of the top countries that contributed to the current study theme is summarized in Table 9, and their world maps are shown in Figure 7.

**Table 9.** Number of documents and citations of the top countries contributing to the research theme.

| Country | No. of Doc | Citations | No. of Doc | Citations |
|---|---|---|---|---|
| | **Scopus** | | **Web of Science** | |
| China | 25 | 161 | 45 | 846 |
| The United States | 14 | 1018 | 25 | 1077 |
| South Korea | 10 | 135 | 15 | 81 |
| Canada | 6 | 964 | 10 | 313 |
| Japan | 2 | 1 | 4 | 130 |
| The United Kingdom | 2 | 8 | 1 | 0 |
| India | 1 | 0 | 1 | 2 |
| Taiwan | - | - | 1 | 2 |
| Australia | 2 | 8 | 2 | 3 |
| The Russian Federation | 2 | 0 | 2 | 130 |
| Singapore | 2 | 17 | 4 | 22 |
| Vietnam | 2 | 8 | 3 | 185 |
| Iran | 2 | 24 | - | - |
| Hong Kong | - | - | 2 | 130 |
| Italy | - | - | 2 | 7 |

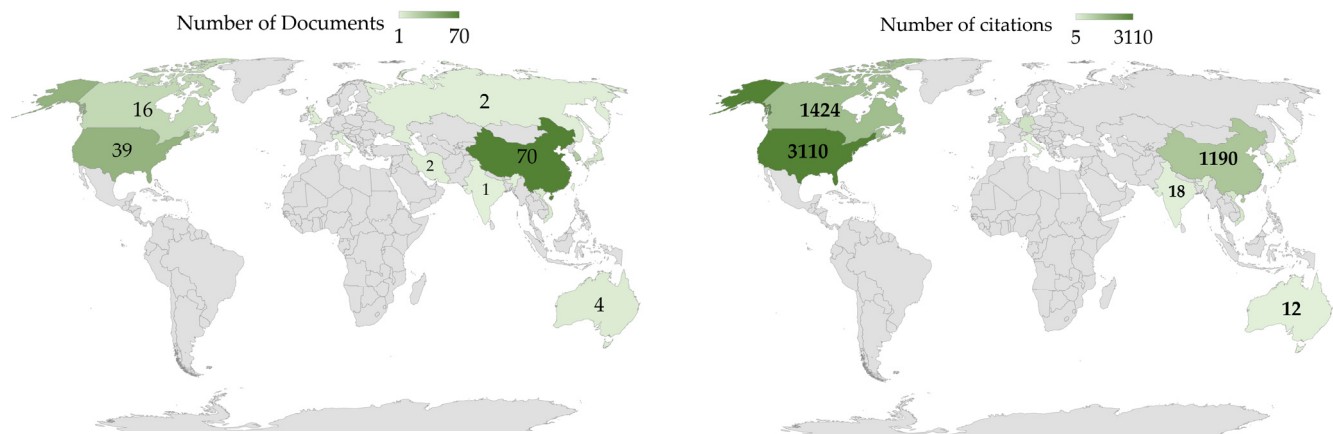

**Figure 7.** Country of origin (No. of documents and citations) analysis.

## 4. DL Based Crack Detection Approaches

In deep learning architectures, CNN is widely used and has shown great performance in crack detection tasks in various civil infrastructures. CNN is a feed-forward deep neural network (DNN) consisting of input, middle, and output layers. The first layer is the input layer, which, in combination with the middle layers, extracts features from the input images and forwards them to the final layer for classification. The architecture of CNN has been evolved, and several architectures are now available as shown in Figure 8.

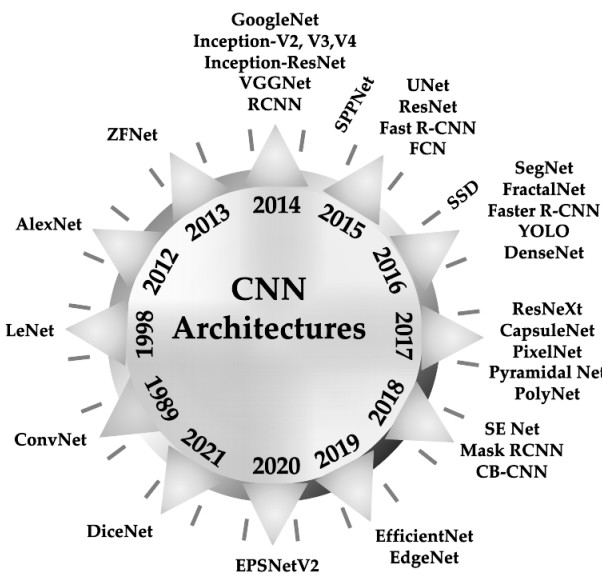

**Figure 8.** CNN architecture evolution.

### 4.1. Crack Detection Using DL Based on Classification and Localization Algorithms

DL algorithms used for the classification and localization of crack are summarized below.

#### 4.1.1. DCNN and Pretrained Approaches

In classification, the DL model extracts useful features and learns crack features from the input images. The crack regions are localized by convolving a sliding window of a fixed size over a high resolution entire image. The image is divided into small patches of specified dimensions, and a bounding box is drawn around the patch classified as a crack patch by the CNN classifier. In the last few years, various efforts have been made to apply the DL model for the classification and localization of cracks in different structures, such as concrete, pavement, and so on. In [20], a CNN-based crack classification algorithm was proposed for crack detection in the bridge surface, using custom data of evenly distributed crack and non-crack 2000 images with a resolution of 96 × 96 and achieved a predictive accuracy of 98.8% on noisy concrete crack images. Dorafshan [21] compared the performance of common edge detectors (Roberts, Prewitt, Sobel, Laplacian of Gaussian, Butterworth, and Gaussian) and CNN for image-based crack detection and localization in concrete structures using 3420 images (319 with cracks and 3101 without). The best edge detector method accurately detected 53–79% of cracked pixels with a residual noise in the final binary images, while the DCNN network accurately detected 86% of cracked images with finer cracks (0.04–0.08 mm). Although the results are not comparable, as edge detection algorithms are pixel-based, while the CNN algorithms are patch-based, the average variation in the F1 score (which is defined as the harmonic mean of the model's precision and recall) of the best deep learning model in comparison with the best edge detector was recorded to be 0.23 and is 1.5 times faster than edge detector models. A CNN model named ChaNet was proposed in [22] for crack identification in concrete structures under lighting conditions using 40,000 images of resolution 256 × 256. The model achieved

96% testing accuracy, 82% precision, 95% recall and 87% F1 score. Comparing the model to the traditional, well-known edge detection methods (i.e., Canny and Sobel [6]), the edge detection methods provided no meaningful crack information and might not be able to treat nonhomogeneous concrete surfaces properly in terms of colour and texture.

The use of a pre-trained network can increase the crack detection capability of a model trained on a small amount of data. A pre-trained DCNN model was trained in [23] for automated pavement crack detection using 1056 images of size 224 × 224. The authors employed and compared specific machine learning, such as neural networks (NN), support vector machine (SVM), random forest (RF), logistic regression (LR), and extremely randomized tree (ERT), and an average variation in F1 score of the TL-NN (transfer learning neural network) classifier in comparison with the other classifier was recorded to be 4.17%; however, the classifier failed to learn to distinguish cracks from joints in PCC-surfaced pavements. Aliyari et al. (2021) [24] compared the performance of various transfer learning (TL) models, namely VGG16 [25], Resnet50 [26], Resnet152-v2, Xception [27], InceptionV3 [28], DenseNet210 [29], NASNetLarge [30], and EfficientNetB4 [31], using 19,023 UAV acquired bridge images. EfficientNetB4 and DenseNet210 achieved the highest accuracy of 96%, followed by ResNet50 and VGG16 with 95% accuracy. Similarly, a comparison of TL methods was performed for the detection of various crack anomalies such as delamination, cracks, spalling, and patches in digital and thermographic images [32]. The MobileNetV2 and VGG16 models outperformed all the compared models and showed average F1 score variations of 7.40% and 11.91%, respectively. Yang et al. (2021) [33] compared the performance of three various deep learning models named AlexNet, VGGNet13, and ResNet18. The ResNet 18 model outperformed the other two neural networks with 98.80% accuracy, 98.90% precision, 98.60% recall, and 98.80% F1 score. A two-step cracks detection and classification architecture by modifying the LeNet-5 model [34] and VGG-16 architecture [25] was proposed in [35]. In the LeNet-5 architecture, the output dimensions of the FC2 (fully connected) layer are modified, and the VGG-16 model is optimized for automatic crack detection of concrete and pavement cracks. The proposed model was evaluated on various publicly available datasets and achieved 91.20% precision, 89.10% recall, and 90.10% F1 score on Cracktree200. Żarski et al. (2021) [36] proposed the TL-based framework KrakN for surface defect detection. The model was trained on 16144 images divided equally into the background and crack categories of size 224 × 224 and achieved 98% accuracy and 97% recall on the KrakN dataset. To improve the accuracy of the crack detection model and reduce the false alarm, the authors in [37] used hybrid (digital and infrared) as an input to the AlexNet architecture. Using hybrid images, the accuracy, precision, and recall of the model were improved by 1.11%, 68.43%, and 9.38%, respectively. Moreover, several TF architectures were used in various works, GoogLeNet [38,39], ResNet-152 [40], Inception V3 [40], Inception V4 [41], and GoogLeNet + ResNet [33,42–46], for crack detection in various civil structures.

In [47], the authors performed optimal hyperparameter selection and trained a DCNN network for crack detection in road structures using 62,408 images with a dimension of 256 × 256. The trained network achieved 99.70% precision, 99.50% recall, 99.60% F1 score, and 7.9 s testing time; in comparison with various ResNet [26] architectures, an average variation of 0.18% variation was observed, while the algorithms were 6–18 times faster than compared algorithms. Liang et al. (2018) [48] also performed hyperparameter selection to address the overfitting problem of various pre-trained networks for system-level failure classification. The VGG16 achieved the highest testing accuracy of 98.98% on a small dataset of 492 images with a resolution of 224 × 224. Similarly, in [49], the authors proposed a crack detection model by adjusting image processing parameters to the optimized value to increase the accuracy of crack detection before using CNN. The proposed model achieved a testing accuracy of 91% using 3000 images for each class. Moreover, in [50], the authors explained the importance of parameter knowledge in addition to model and data knowledge by training a DCNN to detect cracks in the concrete surface, using three publicly available (CCIC [51], SDNET [52], and BCD [53]) datasets. The detection

accuracy of the proposed end-to-end DCNN based transfer learning model was recorded as 99.3% for the CCIC dataset, 99.72% for the BCD dataset, and 97.07% for the SDNET dataset.

The performance of different optimizers for DL-based crack detection was evaluated in [54]. Optimizers play a vital role in the training speed and performance of deep learning models [55]. There are various optimizers available, such as Adam, stochastic gradient descent (SGD), momentum and so on. Based on the comparison, the Adam optimizer showed higher precision with the lowest false-negative rate and highest accuracy. The importance of using the Adam optimizer for a crack detection DL model was also highlighted by the authors in [1]. Rao et al. (2020 [40] evaluate the performance using 15 state-of-the-art convolutional neural network models (Alex Net [56], VGG16 [25], VGG19 [25], ResNet-50 [26], ResNet-101 [26], ResNet-152 [26], InceptionV3 [28], InceptionV4 [57], Inception-ResNet-v2 [57], DenseNet121 [29], DenseNet169 [29], ResNeXt-50-32 × 4d [58], ResNeXt-101-32 × 8d [58], Wide-ResNet-50-2 [59], Wide-ResNet-101-2 [59]) in terms of number of parameters required to train the models, area under the curve, and inference time using 40,000. All the models provide over 95% accuracy and over 87% precision in detecting the cracks. Kim and Cho 2018 [60] used the AlexNet architecture for crack detection in concrete images acquired from an internet search. The softmax layers were modified by developing a probability map to add robustness to the sliding window detection, and a parametric study was performed to determine its threshold. The average accuracy, precision, recall, and elapsed time of the proposed algorithm on various 40 test images were recorded to be 97.02%, 92.36%, 89.28% and 2.02 s, respectively. Kamada and Ichimura 2020 [61] used the restricted Boltzmann machine (Adaptive RBM) for fine-tuning in combination with the deep belief network (Adaptive DBN) for image-based crack detection in concrete structure. The fine-tuning method increased the inference time and accuracy of the model.

Additionally, the importance of training dataset integrity in DL crack classification architecture was shown in [62]. The authors used sampling and training methods based on cross-entropy ranking to address the training class imbalance issue. Three various Alexnet architectures were trained on three various size datasets, namely D1, D2, and D3. The networks trained on a sampled dataset (AlexNet 2 and 3) achieved validation accuracies of 97.46% and 97.55%, respectively, approximately 1.2% higher than that related to AlexNet 1, 96.34%, which indicated that the errors were reduced from 3.64% to 2.54% and 2.45%, respectively. In [63], the authors proposed an external few-shot meta-learning module for handling small and incomplete data size and the internal attribute-based transfer learning module to extract features. The classification accuracy, average precision, recall, and F1 score for all the 10 damage types are 93.5%, 93.1%, 92.9%, and 92.9%, respectively. The training data size, data heterogeneity, network complexity, and the number of epochs have significant effects on the performance of the DL model as studied in [1]. The authors compared a proposed customized CNN model pre-trained networks, i.e., the VGG-16 [25], VGG-19 [25], ResNet-50 [26], and Inception V3 models [28], on eight datasets of different sizes, created from two public datasets. The proposed model is 8–43 times faster than other compared algorithms, and there is no significant change in F1 of the models. The authors showed that the optimal amount of training samples with significant variations has a positive influence on the efficiency of DL-based crack detection models. The effect of data heterogeneity was also studied in [64] using intensity, range and filtered image data types to train deep learning architectures, named Net A, Net B and Net C using 60,000 images with a resolution of 256 × 256. To alleviate the need for post-processing techniques and more training data, the authors in [65] used data augmentation to increase the number of samples to identify the exact location. The proposed method achieved 98.50% accuracy, 1.5% error rate, 98.70% specificity, 13.10% precision, 57.20% recall and 21.40% F1 score. The average variation in the F1 score of the proposed model in comparison with other compared deep learning algorithms was recorded to be 4.32%, which is due to the inclusion of the output function [66] to the encoder. The labeled image data can also be used in integration with other data types, such as fuse 3D LiDAR data images to generate a colorized and semantically labeled 3D map of a structure [67].

### 4.1.2. Modified CNN Architecture

Moreover, DL models can also be used as a feature extractor and integrated with traditional classifiers for crack detection and localization purposes. In [68], the authors used CNN for extracting features from input images and SVM as an alternative classifier to a softmax to enhance model performance. The proposed CNN-SVM model surpasses the CNN model, and an increase of 7.4%, 2%, 10%, and 5% is observed in accuracy, precision, recall, and F1 score, respectively. The same methodology was adopted in [69] for the bridge crack classification algorithm, and the accuracy of the combined model is 91.9% for horizontal, 93.10% for vertical, 91.70% for slope, and 92.30% for block cracks. In comparison with CNN, the proposed model achieved an average variation of 6.76% for various types of cracks. It is still challenging to decide on the architecture and parameters of each model [2]. Individual models may perform well for one classification task but not for another. The authors in [2] introduced a deep learning-based multi-model ensemble approach by combining five different customized convolutional neural networks (CNN) trained on data of 8400 crack and non-crack images with a resolution of 224 × 224.

Furthermore, detecting cracks in other structures, such as tunnels, dams, and masonry, is of vital importance to avoid any unwanted situations. The crack patterns in these structures can be horizontal, vertical, or diagonal. Among all these structures, crack detection in the masonry structure is a tedious task due to the similarity of crack regions with the grout lines. In [39], the authors combined deep learning techniques with traditional image processing algorithms for crack detection, localization, and measurement in tunnels. The authors trained two pre-trained architectures, AlexNet [56] and GoogLeNet [38], on a smaller dataset of 376 images to compare their performance. The AlexNet performed better than GoogLeNet and obtained 93.90% accuracy, 72.40% precision, and 77.80% recall. A ceiling crack detection and localization model, trained, validated, and tested on 1953 ceiling images, was proposed in [70]; 86.22% testing prediction accuracy, 82.53% sensitivity, and 88.94% specificity were achieved. Features visualization was used to demonstrate the classification correctness and provide a localization method for damaged ceiling regions. The authors in [71] combined freeform surface modelling based on the maximum likelihood function and crack analysis based on Mask RCNN for the inspection of tunnel structures using acquired images from a vision measurement unit consisting of camera arrays. Wang et al. (2019) [72] proposed using the Faster R-CNN model based on the ResNet101 framework to detect two categories of damage (efflorescence and spalling) for historic masonry structures. An AP of 99.90% for efflorescence and 90% for spalling damage was achieved by the model, with a 95% mean AP. Sewer pipe defect detection was proposed in [73], using an improved version of a faster region-based convolutional neural network (Faster R-CNN) using 3000 images; 83% mAP and 9.43 FPS speed were achieved. Li et al. (2019) [74] used the YOLO-v3 object detection model for crack detection of dam surface and achieved recall, precision, and F1 score of 0.85, 0.85, and 0.85, respectively.

### 4.1.3. CNN Integrated with Other Modules

In [75], the authors integrated two pivotal attention modules with a very low computational cost in a sequential manner with ResNet architecture to enhance the quality of crack features. The model was trained with 3150 unique crack image samples, with 700 extra images for validation and testing, and achieved an average precision (AP) value of 80.06%. Xu et al. (2019) [53] improved the CNN architecture by introducing the Astrous Spatial Pyramid Pooling (ASPP) module and depth-wise separable convolution to extract multiscale features and reduce computational complexity. The proposed model achieved an accuracy, precision, recall, specificity, and F1 score of 96.37, 78.11, 100, 95.83, and 0.8771, respectively. The model was compared with standard VGG [25] and ResNet [26] and was shown to be 1.5–2 times faster with an average variation of 20.28% in F1 score than the compared algorithms of the crack detection of the dam surface; it achieved recall, precision, and F1 score of 0.85, 0.85, and 0.85.

### 4.1.4. Object Detection Models

Recent strides in DL have enabled the deployment of object detection models for crack detection and localization. The accuracy of these models is measured using mean average precision (mAP), which is calculated by taking the mean of average precision (AP). The AP calculates the detection results of the models by using precision and recall at different threshold levels. The speed of these models is calculated by frames per second (FPS), which is the number of frames that a model processes during a second. Qi et al. (2020) [76] improved the architecture of SSD by removing the network layers that are not useful for prediction and varying the aspect ratio while fixing the short side rather than the area in the prior box setting. The authors used fine labelled 1300 industrial crack images and achieved 81.06% mAP and 52.6 FPS. The proposed model outperformed the traditional SSD and YOLOv3 model since its network is simplified, and priors were designed regarding the crack characteristics. The authors in [77] introduced a receptive field module in the SSD DL framework to enhance the feature extraction capability of the network for real-time pavement crack detection. Similarly, Yan and Zhang [78] proposed a novel network named deformable SSD by adding a deformable convolution to the backbone feature extraction network VGG16. The model was trained and tested on 22,250 crack images of size $562 \times 562$ and achieved an mAP of 85.11%. The precision of the SSD model for road defect detection can be improved by using faster RCNN models [79]. In [80], the authors trained a generic object detection model (an updated version of faster RCNN called RetinaNet) for road damage identification using a quality-aware filter dataset consisting of 18,340 images and 8 various classes. The proposed method was compared with the state-of-the-art model [81,82] and achieved 91.55% accuracy for asphalt structural damages detection from videos and low inference time. The regular convolution and pooling operation with a deformable convolution operation and a deformable pooling operation were improved in three different regular detectors (Faster R-CNN, region-based fully convolutional networks (R-FCN), and feature pyramid network (FPN)-based Faster R-CNN [83].

In Faster RCNN, hyperparameters, such as data size, network depth, number of layers, filter dimensions, or stride values, have a significant effect on the detection accuracy of the model [73,84]. Kalfarisi et al. [85] proposed FRCNN-FED by combining FRCNN with SRFED (structured random forest edge detection) bridge structures using 1250 bridge images of sizes varying from $344 \times 296$ to $1024 \times 796$. The proposed FRCNN-FED model is 10 times faster but less accurate than Mask RCNN. The authors in [86] combined Faster RCNN with 2D digital image correlation (DIC) for concrete crack detection using 1058 images. In the prosed system, Faster RCNN helps in the detection and localization of the crack region, while the DIC helps in the estimation of the deformation field on the concrete surface and measurement of crack width. The architecture was able to localize crack using FRCNN with a certainty of 98%. The accuracy of Faster RCNN was improved by ad hoc YOLOv2 for concrete crack detection [87]. The model was trained on 3010 images of $448 \times 448$ size and is 2.5% more precise, 4.5 times faster in training, and 1.35 times faster in testing time than Faster RCNN on the same data. The accuracy and inference speed of the YOLOv3 model is greater than Faster R-CNN and the SSD network for classifying pavement and concrete distress types [88,89]. Zhang and Chang 2019 [90] proposed an improved version of YOLOv3 for the detection of multiple types of concrete defects. The model was trained on 2206 images ($1280 \times 960$ and $4000 \times 3000$) labelled for four types of concrete damages (crack, pop-out, spalling, and exposed rebar) and achieved 13% better precision than the traditional YOLOv3 model. An improved version of YOLOv3, i.e., YOLOv3-tiny, was proposed in [91] for the crack detection task. The performance of YOLOv3 was compared with the newest versions of YOLO, i.e., YOLO version 4 (YOLOv4) and YOLO version 5 (YOLOv5) for concealed crack detection in asphalt pavement [92]. In terms of processing, YOLOv4 performed better with the fastest detection rate of 10.16 FPS, while in terms of mAP, YOLOv5 achieved the highest mAP of 94.39%. Yao et al. (2021) [93] integrated SPP and PANet modules to the YOLov4 architecture to reduce the number of

parameters for a concrete surface crack detection model. The model accuracy slightly decreased as compared to the traditional YOLOv4 model; however, the number of parameters and inference time of the model decreased. The improved model achieved a mAP of 94.09% with 8.04 M parameters and 0.64 GMacs (Flops). Similarly, the number of parameters of the YOLOv4 model was reduced using the pruning algorithm YOLOv4-FPM. The model was trained on 3487 images of $1000 \times 1000$, $2000 \times 2000$, and $3000 \times 3000$ dimensions and is 20–33 times faster than the traditional YOLOv4 model. Crack segmentation refers to the classification of image pixels into the crack and non-crack pixels.

### 4.2. DL Based Segmentation Algorithms

Crack segmentation refers to the classification of image pixels into crack and non-crack pixels. Various DL methods have been implemented for the detection and segmentation of cracks in various structures [46,48,94–113]. A fully convolutional network (FCN) named Ci-Net trained on two publicly available datasets (CFD [114] and TITS2016 [115] was proposed in [103] for the identification and segmentation of structural cracks. The model achieved 84% precision, 82% recall, 72% IoU, and 60.40% F1 score. Similarly, in [116], FCN was trained for the segmentation of concrete cracks using pre-trained networks as a backbone for the FCN encoders. An automatic pixel-level crack detection and measurement algorithm was proposed in [117] to identify and segment pixel-wise cracks at different scales and measure the morphological features of cracks, providing valuable crack indicators for assessment in practice, such as crack topology, crack length, max-width, and mean width. The architecture of FCN was modified in [118] for the detection of multiple types of damages, such as spalling, efflorescence, cracks, and holes in concrete structures. The architecture was compared with SegNet and the PA, and the MPA of FCN were lower than SegNet, while the FCN achieved higher MIoU and FWIoU than the SegNet.

Moreover, U-net has also been extensively used in crack segmentation task [97,101,102,111,117,119–122]. The authors in [119] used U-net architecture for concrete crack segmentation using a small number of images and achieved the highest IoU of 0.681 at a threshold level of 80. Shokri et al. (2020) [101] also performed crack segmentation in concrete images by training a U-net architecture on a merged dataset consisting of 458 images from CCIC [51] and 670 images from a custom dataset with a resolution of $512 \times 512$. In [102], the authors introduced a self-attention mechanism, truncated expansion strategy, and weighted classification to U-net architecture and compared its performance on various publicly available road datasets, namely CrackTree200 [4], ALE [123], and CFD [114], and an average variation in the F1 score of the model in comparison achieved 38.81%, 51.76%, and 43.66%, respectively. Similarly, a new loss function was introduced by Cheng [104] to the U-net architecture and trained on two publicly available datasets (CFD [114], AigleRN [123]). The average variation in the F1 score of the proposed algorithm in comparison with other considered algorithms was recorded to be 17.58% on the CFD dataset and 8.84% on the AigleRN dataset. SegNet, a novel pixel-wise deep convolutional semantic segmentation network proposed by Zhang [124] has the capability to perform pixel-level classification and has been used in various crack segmentation works [48,112,124–127]. A modified version of SegNet called Seg-DCRF was proposed for crack binarization by combining SegNet and the dense condition random field (DCRF) networks [125]. CrackSegNet for crack detection and segmentation was proposed in [126] with architecture consisting of the backbone network, dilated convolution, spatial pyramid pooling, and skip connection module. The model showed average F1 score variations of 65.41% and 15.37%, in comparison with the traditional method and U-net method, respectively. Recently, Zheng et al. 2021 [127] used the SegNet network for back-end processing for semantic segmentation in a lightweight bridge concrete crack detection model. The proposed model was compared with state-of-the-art algorithms, namely FCN [117], U-net [128], SegNet [129], DeepCrack [130] and CrackU-Net [97] and achieved an average variation in F1 score, recorded to be 5.32%. A deep learning algorithm based on U-Net and a convolutional neural network with alternately updated clique (CliqueNet), called U-CliqueNet, were proposed to separate cracks from a

background in the 60,000 tunnel sub-images with a resolution of 496 × 496 [83]. The model achieved MIoU, precision, recall, and F1 score of 0.869, 0.863, 0.802, and 0.834, respectively. The model was also compared with SegNet [129], U-net [131], FCN [117] and MFCD [132], and an average variation of 6.58% was obtained based on the F1 score.

Zhang [133] proposed an efficient architecture based on the convolutional neural network (CNN) called CrackNet for detecting pavement surface distresses at the pixel level. The authors modified the architecture and proposed CrackNet II [134] to make the model more robust, eliminate local noises and detect fine or hairline cracks. The architecture was further modified, and CrackNet-V was introduced in [135]. CrackNet-V has small filters in the convolutional layers for reducing the number of parameters and a new activation function called "Leaky Rectified Tanh" for accurate detection. CrackNet-V only takes 0.33 s on average for pixel-level crack detection, which is roughly 3–4 times faster than CrackNet. In [136], the author's proposed a three-phase architecture for crack detection and segmentation in concrete structures. The first and second phases use 2D CNNs for the classification and segmentation of cracks. In the third phase, the crack length and width are estimated. The crack classification algorithm achieved an accuracy of 98.98%, and the segmentation network achieved an accuracy of 100% with an IoU value of 0.87. The model can measure crack length and width of 401.33 mm and 2.2 mm, respectively. Similarly, a two-stage DL architecture was used for the segmentation and identification of distress types on road surfaces using images acquired from a UAV [137]. In the defect segmentation task, the VGG model outperformed the other considered models and achieved IoU of 89.2, 66.2, and 85.7 in detecting path holes, rutting, and wash-boarding. Pan et al. (2020) [94] proposed a pixel-level crack detection and segmentation system by using a hybrid deep learning approach called SCHNet (spatial–channel hierarchical networks) on a portion of publicly available datasets [24]. The average variations in the means IoU of the proposed algorithm on the test data in comparison with U-Net [131], Deep-Labv3 [138], PSPNet [139], and Ding [140] is 15.48%. The authors in [95] created a dataset named EdmCrack600 consisting of 600 annotated images of 1920 × 1080 resolution and a DL-based new algorithm ConnCrack by combining the conditional Wasserstein generative adversarial network with connectivity maps for crack detection and the segmentation of road cracks. The average variation of the algorithm in comparison with other machine learning approaches on CFD [114] and the proposed EdmCrack600 dataset was recorded to be 9.37% and 33.67%. The proposed algorithm is 1.2–4 times faster than other compared algorithms.

Crack segmentation based on object detection models have been performed in various studies [107,110,111,141]. Zhang et al. [98] added semantic segmentation to the Mask RCNN model to extract more information about the crack to improve the prediction accuracy of the model. The architecture exhibits 13.18%, 4.52%, variations in the mask AP and AP box on GDPH (Guangdong Provincial Highway) custom dataset in comparison with Mask RCNN and PANet (Path Aggregation Network). The authors in [142] combined Faster region proposal convolutional neural network (Faster R-CNN), modified tubularity flow field (TuFF), and modified distance transform method (DTM) crack detection, segmentation, and quantification. The hybrid network achieved an average precision of 95% in crack detection, 83% IoU in crack segmentation, and 93% accuracy in the crack quantification process for crack length and thickness of a 2.6-pixel root mean square error. Majifarid et al. 2020 [143] also creates a hybrid model by combining YOLO and U-net architecture for classifying and segmenting nine road distress types using a dataset consisting of 7237 images extracted from Google Street-view. Various pre-trained architectures are used as encoders for U-net [131], FCN [117], FPN [144], a generic pyramid representation to perform pixel-level crack segmentation in masonry structures [145]. The average variation of the combined models i.e., U-net-MobileNet and FPN-InceptionV3 in comparison with other networks is recorded to be 2.85%. Crack detection and segmentation algorithm (CDDS is proposed) for dam surface in [146] using images collected from a drone. The proposed network achieved recall, precision, F1 score, Crack IoU, and Background of 80.45, 80.31, 79.16, 66.76, and

99.76. On comparison with SegNet [129], Unet [131], FCN [117] and ResNet152 [147], an average variation of 9.53% is observed in F1 score due to integration multi-level features.

## 5. Analysis and Discussion

The analysis of the retrieved articles is performed based on the dataset, its size, the domain in which the study was performed, the architecture and precision of the model, as explained in detail below.

### 5.1. Dataset Based Analysis

Before the development of a vision-based crack detection system, dataset acquisition is performed for training the models. The selection of a feasible dataset for model training remains a great challenge that needs to be resolved. In the retrieved articles, 72.12% of articles are based on a custom dataset among which 64.71% used less and 10K images for model training while the remaining used more than 10K images. The custom dataset is preferred by most researchers due to the lack of a standard dataset for crack detection tasks and enables them to capture specific requirements of the crack detection algorithm being used. In the custom dataset-based articles, 62.73% of studies are conducted for concrete crack detection while 28.87% are performed for crack detection in pavement surfaces. It can be inferred that the majority of the custom datasets consisting of less than 10K image samples are used for training concrete crack detection models. In 8.40% of articles, the researchers built their datasets for crack detection in different types of materials namely dams, glasses, masonry, ceiling, rock cliffs, and nuclear plants.

Moreover, 17.57% of the retrieved studies are based on publicly available datasets in which 74.41% of datasets consists of more than 10K images while 25.59% of the public dataset have less than 10K images. The pre-built datasets are available online and are used as a standard for the training and testing of the crack detection models. Among these studies, concrete crack detection is performed in 58.13% of articles while pavement crack detection is performed in 41.85% of studies. The analytics shows that public datasets such as CFD [114], AigleRN [123], Crack500 [148], AEL [148] are frequently used for pavement crack detection while for concrete crack detection SDNet, CCIC [51], and Özgenel's [51] are commonly used for crack detection. From the above analysis, it can be inferred that there is no public crack detection dataset available for other structures such as masonry, dams, sewage pipes, tunnels, and so on. Therefore, this points towards the need to create a publicly available dataset for such structures to solve the inspection problem efficiently. The overall distribution of the articles is shown in Figure 9 and the articles are shown in Table 10.

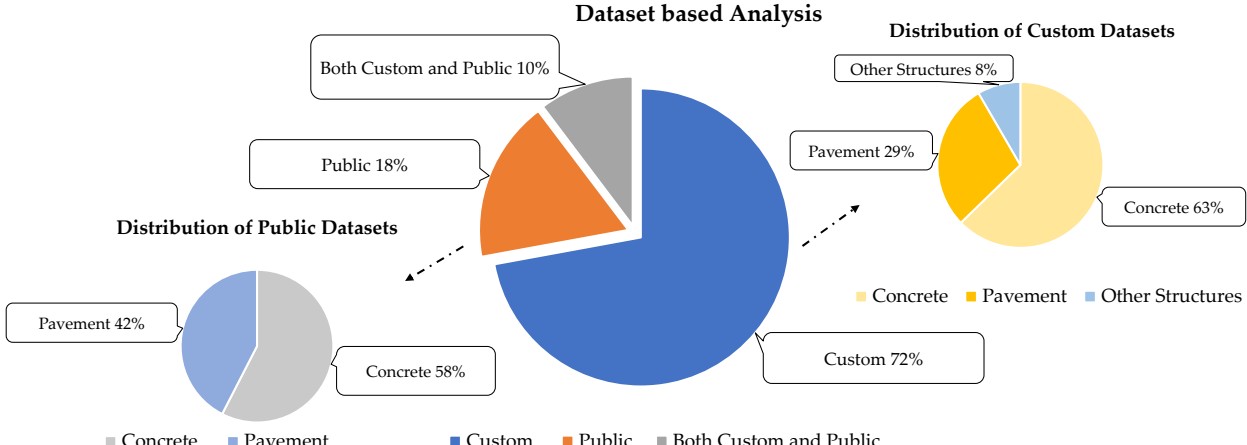

**Figure 9.** Distribution of retrieved articles based on the dataset.

**Table 10.** Image datasets on crack detection.

| Dataset Type | Domain | Works |
|---|---|---|
| Custom | Concrete | [2,20–22,32,33,36–39,43–45,48,49,60,62,63,67,69,71,73,75,84,85,87,89–91,93,97,105,107,108,110,112,117–119,121,126,127,130,133,141,142,146,149–171] |
| | Pavement | [20,40–42,47,65,77–80,88,92,100,109,113,125,131,133,135,137,143,172–183] |
| | Other structures | [46,54,68,70,74,76,145,164,184,185] |

| | Domain | Dataset Name | Works |
|---|---|---|---|
| Public | Concrete | FCD | [161] |
| | | CCIC' | [1,106,136,186] |
| | | Xu's dataset | [187] |
| | | DIV2K | [188] |
| | | SDNET | [35,60,65,97,119,189,190] |
| | | Özgenel's | [65,97,119,189,191,192] |
| | | KrakNet | [69,100,133,190,193,194] |
| | | CCIC + SDNET + BCD | [50] |
| | | CFD + TITS2018 | [103] |
| | | CFD + Online search | [195] |
| | | SDNET + CCIC | [1] |
| | | LTPP + GAPS + APRDC | [196] |
| | | SDNET + Özgenel's | [189] |
| | | KITTI + Cityscapes + CFD | [111] |
| | | CFD + AigleRN | [197] |
| | Pavement | AEL | [123,190] |
| | | Crack500 | [120,190,193] |
| | | CFTD | [124] |
| | | CLS | [96] |
| | | CFD Llamas | [103,198] |
| | | Stone331/GAPs384 | [193] |
| | | TRIMMD | [124] |
| | | CCD861 | [35] |
| | | CrackTree260 | [193] |
| | | EdmCrack1000 | [194] |
| | | Cracktree200 | [35,102,190,199,200] |
| | | EdmCrack600 | [95] |
| | | CFD | [35,44,99,102,124,151,193,194,200–203] |
| | | AigleRN | [27,40,49,104,203] |
| | | MCD | [204] |
| | | AIMCrack | [199] |
| | | CCD1500/CCD861/CCD861 | [35] |
| | | DeepCrack | [35] |
| | | PCD19/CRKWH1000/ | [193] |

| Dataset Type | Domain | Works |
|---|---|---|
| Custom and Public | Concrete and Pavement | [24,35,83,97,98,101,164,168,170,188,197,200,205–209] |

### 5.2. Domain-Based Analysis

In the reviewed articles, every study is conducted to perform crack detection in a specific civil structure as depicted in Table 11. The majority of the articles are focused on detecting cracks in concrete and pavement and contribute approximately 85% of the reviewed articles as shown in Figure 10.

The reason is that concrete and pavement cover many infrastructures elements, such as roads, bridges, buildings, walls, and so on. Several studies are present on crack detection in other structures namely tunnels, dams, sewage pipes, masonry structures, industry, ceilings, nuclear plant, glass, dam, and rock cliffs, which cover 15% of the articles reviewed in the current study. Hence, there is a need to focus on developing crack detection algorithms for these structures also.

**Table 11.** Published articles on crack detection (domain wise).

| Area/Domain | Number of Works | References |
|---|---|---|
| Pavement/Roads | 59 | [23,35,41,42,47,64,65,77–80,88,92,95,96,98–100,102,103,109,111,113,120,125,130,133–135,137,143,168,172–183,190,193–196,198–203,205,207,208] |
| Concrete | 65 | [1,2,22,24,32,33,37,38,40,43–45,50,53,60–62,75,80,83,86,87,91,93,97,101,106–108,110,112,116–119,121,124,130,133,135,136,141,142,146,150,152–154,156,158,161,164,166,167,169,171,186,189–192,197,204,206,209–212] |
| Bridges | 18 | [20,21,36,48,63,65,67,69,85,89,90,94,127,149,162,170,187,188] |
| Sewage Pipes | 5 | [73,75,84,159,163] |
| Tunnels | 9 | [39,41,71,105,126,155,157,160,165] |
| Masonry | 4 | [68,72,145,185] |
| Industry/Ceiling/ Nuclear Plant/ Glass/Dam/Rock Cliffs | 6 | [46,59,70,76,84,184] |

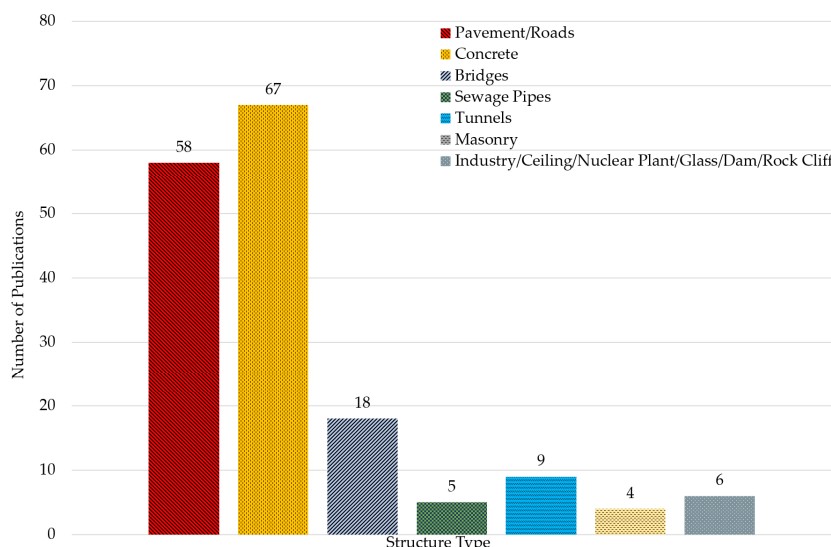

**Figure 10.** Distribution of retrieved articles based on domain.

### 5.3. Architecture Based Analysis

Each article reviewed in this work is based on a specific DL architecture. As summarized in Table 12, DCNN is mostly used for crack classification in various structures followed by CNN TL models, such as MobileNetV2, AlexNet, VGG-16, GoogLeNet, and so on. Performing crack detection using CNN, some of the authors have trained the model from scratch, while most of them opted for pre-trained models because of their capability to achieve better crack classification performance on large training datasets. The remaining CNN-based architectures are either hybrid, or some modules are integrated with them. In the object recognition-based DL models, Faster RCNN and YOLO are the most implemented architectures for the recognition and localization of the crack regions. The reason behind the abundant use of FRCNN is the architecture's ability to localize small cracks. However, these approaches have the limitation of treating the whole crack as a single object. These models are also integrated with other modules for crack segmentation tasks to locate the crack region more precisely [85,142]. In crack segmentation, Unet is the most adopted DL learning architecture in the reviewed literature followed by the SegNet model. The segmentation of cracks is a tedious task and require a powerful architecture to accurately segment cracks in the images. U-net has shown promising performance in various applications, including crack segmentation. In comparison with

other segmentation networks, the encoder–decoders network architecture (U-net) requires a small amount of data to train and provide crack location information precisely. Various authors have also improved state-of-the-art DL models and named their architectures as APLCNet [98], RetinaNet [80], [172], SCHNet [94], Ci-Net [103], Cls-GAP Net [96], EfficientNet [168], ProposedNet [109], CMDNet [161], TernausNet [185], Spiral Net [166], LDCC-Net [189], RRCE-Net [170], RAO-UNet [181] and so on. From the above analysis, it can be inferred that crack segmentation can be performed by implementing a simple lightweight encoder–decoder network that can be trained with less data and resources.

**Table 12.** Published articles on crack detection (domain wise).

| Architecture | Works | Precision |
|---|---|---|
| DenseNet201 | [24] | NA |
| LeNet-5 + VGG16 | [35] | 91.20 |
| KrakN | [36] | NA |
| Inception V3/Inception V4 | [40]/[41] | NA/90.00 |
| GoogLeNet/GoogLeNet + CDN/GoogLeNet + ResNet | [38,39]/[99]/[45] | 98.72, 35.30/84.35/93.50 |
| ResNet/ResNet-152 | [33,40,42–44]/[46] | 98.90, NA, NA, NA, NA/90.0 |
| DBN | [61] | NA |
| R-FCN | [83] | 90.01 |
| 2D DIC + FRCNN | [86] | NA |
| YOLOv5x | [92] | 74.00 |
| ConnCrack | [95] | 96.79 |
| DenseCrack | [99] | 92.17 |
| RPN | [105] | NA |
| FCN | [46,100,105,106,116,117] | NA, 98.33, NA, 91.30, 89.30, 97.60 |
| U-Net/UNet-EB | [97,101,102,104,111,119,121,122]/[120] | 98.23, NA, 87.48, 40.44, 92.12, 96.52, NA, NA/57.50 |
| SegNet Bottleneck Depth-Separable Convolution With Residuals | [127] | 97.95 |
| Deep Crack | [130] | 86.60 |
| YOLO/YOLO + U-Net | [87]/[143] | 77.00/93.00 |
| CDDS | [146] | 80.31 |
| CrackNet/CrackNet II/CrackNet-v/CrackPix | [133]/[134]/[135]/[151] | 90.13/90.20/92.58/92.55 |
| SSD/SSDLite-MobileNetV2 | [76–79]/[154] | 81.06, 78.69, 19.45, 95.05/52.00 |
| ANet-FSM | [156] | 99.10 |
| MobileNet-v1 + ASPP | [160] | NA |
| YOLO-v3/YOLOv3-tiny | [33,74,88–90]/[91,92,163] | NA, 85.00, 73.64, 76.50, 66.20/98.00, 64.00, 85.37 |
| SegNet/CrackSegNet/SegNet-DCRF/FL-SegNet | [48,112,124]/[126]/[125]/[164] | , NA, NA, 82.0 /63.85/97.26 |
| Multilayer ELM-based crack detector MECD | [167] | 97.80 |
| RetinaNet/ SCHNet/ Cls-GAP Net APLCNet/Ci-Net/ ProposedNet /EfficientNet. | [80,172]/[94]/[96]/[98]/[103]/[109] [168] | NA, 85.80/NA/93.20/92.20/NA/ NA/82.30 |
| ResNet18 + YOLO_v2 | [171] | 89.00 |
| Edge detection + CNN-CDM | [176] | 1.00 |
| CAE | [178] | NA |
| FCN + GCRF + Uncertainty framework + probability-based rejection net | [179] | NA |
| RNN | [180] | NA |
| CNN (MobileNetV2)/CNN (AlexNet)/CNN (VGG-16)/ CNN (VGG-19) | [32]/[33,37,39,60,62,192]/[32,33,46,48]/[40,181] | 66.70/97.60, 98.72, 72.40, 92.36, 92.0, 99.0/55.60, 98.60, NA, NA, NA/90.0, 1.00 |
| PointCrack3D | [184] | 5.00 |
| Faster RCNN | [48,72,73,79,83,84,153,158,174,186] | 80.00, 95.00, 91.58, 26.45, 83.0 83.0, NA, 87.80, 66.89, NA, |
| SSENets/CrackU-Net/U-CliqueNet | [155]/[175]/[187] | 86.30/NA/95.45 |
| SrcNet | [188] | NA |
| CMDNet/TernausNet/Spiral Net/LDCC-Net/RRCE-Net/RAO-UNet | [161]/[166]/[170]/[181]/[185]/[189] | 98.90/78.44/99.28/98.32/81.90/NA |
| Xception | [191] | NA |
| CCapFPN | [193] | 91.40 |
| DCNN | [20,21,23,39,47,49,50,54,62–65,67,70,75,136,137,162,165,180,182,183,194] | NA, 97.90, 90.0, 88.30, 99.70, NA, NA, 89.60, 92.0, 93.10, 99.60, 13.10, NA, NA, 80.60, NA, NA, NA, NA, NA, NA, 91.00. |
| Mask R-CNN using Detectron2′s /Mask RCNN | [113]/[71,85,107,110,111,141,173,195] | 95.00/ NA, 66.0, 90.40, 90.0, NA, NA, 96.32, NA |
| YOLOV4/YOLOV4-FPM | [93,149]/[196] | 94.09, 97.60/78.00 |
| CCA + CRA | [197] | 83.40 |
| Encoder-decoder + SWM | [199] | 67.20 |
| ICGA | [198] | 89.33 |
| SSGN | [200] | 33.55 |
| U-HDN | [201] | 94.50 |
| Deep Residual Net | [202] | 93.57 |
| CNN + ASPP CDDS / CNN + SVM/ CNN with U-Net and FPN/ CNN with PAN | [53,69,145,204] | 78.11, NA, 95.30, 81.30 |
| RCNN + CNN | [205] | NA |
| GAN | [207,208] | 89.70, NA |
| FRCNN + FPN/FRCNN + TuFF + DTM/FRCNN-FED/FRCNN-Inception-ResNet-v2 | [83]/[85]/[142]/[209] | 92.11/78.00/95.00/60.80 |
| Customized CNN | [1,2,22,68,177,190,210] | 97.30, 99.60, 98.90, 82.00, NA, NA. |
| MALSSS | [211] | 98.17 |
| FF-BLS | [212] | 99.90 |

## 5.4. Precision Based Analysis

The effectiveness of a crack dataset can be assessed by reviewing its performance outcomes for crack detection. In the reviewed studies, researchers used various evaluation metrics, such as accuracy, precision, recall, and F1 score for the validation of their proposed crack detection systems. In the proposed articles, precision is considered for dataset-based performance analysis, as it is mostly used in the reviewed articles. The precision values of studies based on the custom dataset are summarized in the table. The majority of concrete and pavement surfaces achieved a precision of more than 75%; however, the articles using less than 10,000 images achieved more precision than articles based on a dataset consisting of more than 10,000 images. The distribution of the articles based on precision is depicted in Table 13. The precision based comparative analysis of the algorithms is not provided in the paper, as the majority of the works are based on a custom dataset, and it is impossible to provide a fair comparison of the algorithms unless all the models are evaluated in a unified setup.

**Table 13.** Distribution of the published articles on crack detection (precision wise).

| Structure | Dataset Size | Precision (%) | References |
|---|---|---|---|
| Concrete | <10,000 | 95–100 | [2,97,127,142,149,156,161,170,187] |
| | | 90–95 | [63,83,91,98,106,107,110,150] |
| | | 85–90 | [74,89,101,157,158,163,171,207] |
| | | 80–85 | [48,73,75,84,106,117,130,146,168,197] |
| | | 75–80 | [53,85,87,164,206] |
| | | <75 | [32,39,90,126,154,169] |
| Concrete | >10,000 | 95–100 | [1,33,37,38,121,167,192,212] |
| | | 90–95 | [21,35,45,60,62,93,116,151] |
| | | 85–90 | [103,204] |
| | | 80–85 | [22,152,165] |
| | | 75–80 | [178] |
| | | <75 | NA |
| Pavement | <10,000 | 95–100 | [95,113,175,176,181,193,201,203] |
| | | 90–95 | [23,35,92,134,135,143,194,199,201] |
| | | 85–90 | [130,133] |
| | | 80–85 | [124,135,174,178] |
| | | 75–80 | [77,196] |
| | | <75 | [65,79,102,120,123,199,200] |
| Pavement | >10,000 | 95–100 | [47,64,78,100,125,173] |
| | | 90–95 | [40,180] |
| | | 85–90 | NA |
| | | 80–85 | NA |
| | | 75–80 | NA |
| | | <75 | [88] |
| Other structures | < & >10,000 | 95–100 | [72,185] |
| | | 90–95 | [96] |
| | | 85–90 | NA |
| | | 80–85 | [68] |
| | | 75–80 | NA |
| | | <75 | [184] |

## 6. Gaps, Challenges, and Future Research Opportunities

Crack detection using DL approaches has seen exponential growth over the past few years. Since cracks are irregular in form with no specific shape or size, it therefore becomes difficult for a DL to perform a crack recognition task. Additionally, the considered domain poses further challenges that need to be addressed to enable the DL algorithms to replicate human perceptions.

### 6.1. Automatic Recognition of Crack Type and Severity

The current approaches focus on the detection and segmentation of cracks; however, there is limited literature available focusing on the nature and severity of cracks, concerning various structural components. A few of the available studies based on crack width and topology considered in this review article are [117,119,125,136,150,154,155,183,184]. To the best of the author's knowledge, crack types, namely meandering and crescent-shaped, have not yet been considered in any article. It is therefore imperative to carry out research studies that consider classification and other factors to make the task of crack detection more efficient. Moreover, efforts have been put forth to convert crack pixel values to a physical length and width [117,213–215], which can be calculated by using a conversion scale (mm/pixel). To convert the crack width from pixel units to millimeters, the camera sensor's physical size (in millimeters) and image resolution (in pixels) are required [216]. However, the transition of pixels to geometric properties still needs a lot of improvement.

### 6.2. Dataset Availability and Suitability

From the review, the majority of the DL-based crack detection models are based on custom datasets consisting of small image patches acquired in a close range using a digital camera [36,172], moving vehicle with a mounted camera [71,95,173,180], smartphone [81,99] and UAV [24,119,154]. There are limited studies available on crack detection using composite view of the structure by creating a continuous mosaic. To provide a global view of the crack density of the structures, the drone, aerial and satellite images in combination with close-range images must be taken into consideration to provide a better view of the structure's cracks to the inspectors. In the areas where image acquisition cannot be performed by using a drone or handheld cameras due to high traffic flow or other environmental factors, aerial and satellite images can provide a cost-effective solution.

The performance of the DL model depends on the various factors of the input data, such as variation among the samples, textures, noises, samples resolution, illumination, and so on. Additionally, there is a lack of a public dataset for structures such as masonry, dam, etc. Therefore, it is necessary to make a standard dataset from various civil structures. The dataset must ensure variation among the data samples, and the data collection should be performed in various lighting conditions to validate the performance of various DL-based approaches. The dataset must be made publicly available to the research community with specified samples in the training, cross-validation, and testing, which in return will enable researchers to validate their approaches on a single data for a fair comparison. In situations where the data availability is limited, data augmentation approaches can be used to increase the dataset samples to achieve better performance; however, the selection of the data augmentation approach is still not considered in the literature. Similarly, the need for larger datasets can also be addressed by using ensemble modelling.

### 6.3. Efficient Data Preprocessing Techniques

In addition to data, researchers use various preprocessing approaches to address the presence of noises, blemishes, and shadows in the images while performing crack detection using DL techniques; however, more efforts are required to build automatic and efficient algorithms to address the removal of noise and irregularities from the images before being given to DL models. Additionally, sometimes the quality of images acquired from UAVs is adversely affected by the camera vibrations due to inadequate operator skills and atmospheric factors, such as wind direction and speed [217]. The blur caused

due to the moving camera reduces the image sharpness and makes the crack detection task difficult. Various DL-based deblurring approaches have been proposed in the past to address the issue in various applications [218–221]; therefore, it can also be considered to deblur the images containing cracks to improve the detection accuracy.

### 6.4. Automatic Labelling Approaches for Crack Detection

Currently, very little effort is made to perform crack segmentation using a weakly supervised or unsupervised manner. In the supervised segmentation approaches, the dataset labelling for crack detection is performed manually, which is laborious and time-consuming. To minimize these human interventions, either the crack segmentation task should be performed in an unsupervised manner or an automatic labelling method, which can automatically perform the manual labelling of the cracks. The speed of the CNN-based crack detection methods highly depends on various factors, such as the selection of hyperparameters, network depth, and fine-tuning of the architecture.

### 6.5. Parameter Tuning and Optimization

In the review, most of the algorithms adopted to noise and image variations have experienced longer processing time; therefore, it is required to necessarily explore the network depth and perform parameter tuning to achieve better performance with less computational time and cost. Moreover, numerous approaches have demonstrated optimization techniques for the fast convergence of deep learning models with minimum iterations. To the best of the author's knowledge, there is a lack of studies that take the selection and combination of various optimization algorithms to help DL models. Therefore, there is a serious need to perform studies on different optimizers and their combination (hybrid optimizers) and examine the strength and weaknesses of each optimizer to set the stage for future research.

### 6.6. Integrating New Features beyond RGB Images

The last few years have also seen various DL-based crack detection studies using 3D images [222,223]. The state-of-the-art DL approaches can also be improved to adapt the 3D imaging technologies to improve model generalization ability and consider new crack features, such as crack dimensions. Finding the crack dimensions will not only help in determining the severity of the crack but will also help the authorities in deciding on the structure usage. Crack detection focuses on capturing the current state of the structures using vision-based techniques. Regression models may help to predict and quantify the number of cracks on structures using multiple features, such as (a) time since last maintenance, (b) type of last maintenance (i.e., surface or major), (c) environmental factors (freeze-thaw cycles, temperature, depth of water table beneath, humidity and precipitation [224,225]), (d) deterioration measure (road rutting [226], international roughness index (IRI) [226], deflection and pavement condition index (PCI [227])), and (e) traffic factors (lane, traffic volume) [228]. The advantage of data fusion is two-fold: (a) to predict the progression of cracks within the coming years and (b) to improve the accuracy of the state-of-the-art crack detection models. Further research is needed to assess the advantage of blending these features to improve the accuracy of RGB-based automatic crack quantification methods.

### 6.7. Realtime Crack Detection

Additionally, the DL model does not perform well in real-time scenarios as the model is trained offline. There is a need to address the issue by proposing a robust model and evaluating the model in real-time under various environmental conditions.

### 6.8. Improving the Pixel Level Segmentation Accuracy

For the semantic segmentation task, U-net and FCN are widely used in the literature for crack classification at the pixel level. However, the segmentation of hairline crack

remains a challenge for the researchers, and straight networks usually require more time to train. The improvement in the architecture of U-net, such as removing pooling layers and introducing objective and loss functions, might enhance the performance and reduce the computational cost of the crack segmentation task. The segmentation is also performed by integrating CNN with other modules, such as crack delineation network (CDN), pyramid attention network (PAN), and so on. However, it still requires more attention from the researcher working in the field to increase the accuracy of crack detection at the pixel level. The DL model consists of millions of parameters with large memory and high computing devices; therefore, implementing DL for various problem challenges still requires research.

## 7. Conclusions

This paper provides an extensive review of deep learning-based crack detection approaches developed in the last 11 years and published in top-quality journals and conferences. A detailed review of 165 articles retrieved from Scopus and WoS after applying five different criteria is performed. A detailed summary of the developments of CNN-based structural crack detection architectures is presented in this paper. The analysis of the articles is performed based on the used dataset, its characteristics, algorithms, their performances, and application domain. The paper also provides a discussion on the gaps, challenges, and future research opportunities in the field of crack detection using deep learning approaches. From the analysis, it can be inferred that most researchers prefer to use custom datasets when it comes to crack detection since there are no standard datasets available. Moreover, there is limited work available on structures, such as masonry, dams, sewage pipes, tunnels, etc. As a result, an efficient algorithm and open dataset for such structures would solve the inspection problem more efficiently. It can also be inferred that collecting an optimal amount of data at various lightning conditions and providing variation among data samples can increase the performance of deep learning models. For concrete and pavement surfaces, more than 75% precision was achieved in the reviewed articles; however, those using datasets containing fewer than 10,000 images achieved more precision than those using datasets containing more than 10,000 images.

In the reviewed articles, CNN is widely used for crack detection tasks and its performance heavily relies on various factors, such as the selection of hyperparameters and fine-tuning of the architecture. For the crack localization, sliding window techniques have shown promising results; however, it is difficult for the sliding window approach to localize crack pixels. It is possible to segment fine cracks using crack segmentation algorithms, such as U-net. Moreover, the U-net architecture can be enhanced for improving the performance and reducing the computational cost of crack segmentation, removing pooling layers and introducing loss functions. Another observation is that the crack property, nature, and severity for the diverse structural components should be considered while performing crack detection using DL approaches. Using 3D imaging techniques, the ability to generalize the model and find new crack features, such as crack depth, can be enhanced. The articles also highlight various research gaps and challenges that need to be addressed, such as common publicly available datasets having specified training, validation and testing samples, automatic data labelling software, pre-processing methods for handling noises and irregularities, and so on. Overall, the proposed review provides a solid knowledge of various DL-based crack detection approaches and will work as benchmarks for researchers working in the field to assist them. The limitations of the proposed study are that (a) articles written in the English language were chosen from both libraries, while papers written in any other language were excluded; (b) to cover the majority of the related articles and minimize the bias caused by the keyword search, most commonly used keywords were included, but some research studies may have used synonyms that were not included in the keyword search; however, we think this risk is low because of the adapted methodology. In the future, the authors are planning to include different comparison criteria, such as the inference time of algorithms, network depth, and implementation, for reviewing DL-based crack detection approaches.

**Author Contributions:** Conceptualization, L.A., F.A. and H.A.J.; methodology, L.A., W.K., F.A. and H.A.J.; writing—original draft preparation, L.A., H.A.J. and F.A.; writing—review and editing, F.A., H.A.J., W.K., M.A.S. and L.A.; supervision, F.A., H.A.J. and M.A.S. All authors have read and agreed to the published version of the manuscript.

**Funding:** This research received no external funding.

**Conflicts of Interest:** The authors declare no conflict of interest.

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
