# Peer review of "Bibliometric Analysis and Review of Deep Learning-Based Crack Detection Literature Published between 2010 and 2022"

_buildings, doi:10.3390/buildings12040432_

Round 1
Reviewer 1 Report
The purpose of this paper is to conduct a bibliometric analysis and review of Deep Learning-based Crack Detection literature published almost in the past decade. The authors conducted good research, and the paper is well-written. After addressing some corrections, it can be published.
- Please use appropriate references for lines 36 to 46.
- A good research methodology has been conducted; only please clarify how the main keywords have been selected. Based on the knowledge of the authors? If yes, it may cause bias in the research,. Please refer to appropriate references.
- A table for the inclusion and exclusion criteria should be presented in this paper; for example, see: https://doi.org/10.1016/j.ijdrr.2021.102627
- Limitations of the study should be discussed in conclusion.
Author Response
we thank the reviewer for his/her valuable and encouraging comments/feedbacks and for. We are delighted to resubmit the revised manuscript. The files uploaded are:
(a) a point-by-point response to the comments raised by the respected reviewers
(b) a revised version of the manuscript.
We look forward to hearing from you at your earliest convenience.
Thank you for your time.

Reviewer 2 Report
The author only simply summarized the deep learning-based crack detection-related articles in the current version. The authors need to significantly improve this literature review using the following suggestions: (a) Missing leading journals. In table 2, rank 1 and rank 14 are the same journal. The authors skipped all ASCE Journals of Computing in civil engineering, Performance of constructed facilities, Transportation engineering Part B pavements. https://ascelibrary.org/action/doSearch?field1=AllField&AllField=crack%2C+deep+learning&ConceptID= (b) Add precision based analysis. The authors can consider the Comparative performance analysis presented in [1]. (c) The reviewed articles did not explain how to use the crack detection results. In addition, the authors should think more about How to obtain the images (i.e., drones, vehicles, satellites)? Do the images have a scale for converting pixel to physical length. An example study can be found in [2]. In addition, the crack image deblurring approach should be discussed if images are captured with camera movement. (d) The reviewed articles only use RGB images as inputs. It would help if you could discuss using multiple features (e.g., RGB, elevation) as the input data, see the example study in [3]. [1] "Automated digital modeling of existing buildings: A review of visual object recognition methods," Autom. Constr., vol. 113, p. 103131, May 2020, doi: 10.1016/j.autcon.2020.103131. [2]"Development of a Pavement Evaluation Tool Using Aerial Imagery and Deep Learning," J. Transp. Eng. Part B Pavements, vol. 147, no. 3, p. 04021027, Sep. 2021, doi: 10.1061/JPEODX.0000282. [3]"Identifying Asphalt Pavement Distress Using UAV LiDAR Point Cloud Data and Random Forest Classification," ISPRS Int. J. Geo-Information, vol. 8, no. 1, p. 39, Jan. 2019, doi: 10.3390/ijgi8010039.Author Response
we thank the reviewer for his/her valuable and encouraging comments/feedbacks and for. We are delighted to resubmit the revised manuscript. The files uploaded are:
(a) a point-by-point response to the comments raised by the respected reviewers
(b) a revised version of the manuscript.
We look forward to hearing from you at your earliest convenience.
Thank you for your time.

Reviewer 3 Report
The paper is a state-of-art review on deep learning methods applied in identifying the crack pattern of civil structures. The topic is very actual and of interest, therefore it should be published in buildings. The paper is well organized and structured, some minor revisions on my side can be found in the attachment. Just few curiosities to ask to the authors:
- How many papers assess the crack width? On how many specimens? How is the reference crack width evaluated?
- What is the best software to employ for the deep learning-based crack detection? Matlab? Python?
- In your analysis, did you find also papers concerning deep learning-based area detection (i.e. for detachments)? How many papers?
Are the employed CNN/DL models already available routine? Therefore can they used in software such as matlab or python to perform other analyses?

Author Response

(The authors gave the same response as above.)

Round 2
Reviewer 2 Report
All my comments have been addressed, Thank you!